# Recognition of Plastic Film in Terrain-Fragmented Areas Based on Drone Visible Light Images

**Xiaoyi Du [1], Denghong Huang [1,2,*], Li Dai [1] and Xiandan Du [1]**

1 School of Geography & Environmental Science, School of Karst Science, Guizhou Normal University, Guiyang 550001, China; 232100090358@gznu.edu.cn (X.D.)
2 State Engineering Technology Institute for Karst Desertification Control, Guiyang 550001, China
* Correspondence: hdh@gznu.edu.cn; Tel.: +86-13658515030

**Abstract:** In order to meet the growing demand for food and achieve food security development goals, contemporary agriculture increasingly depends on plastic coverings such as agricultural plastic films. The remote sensing-based identification of these plastic films has gradually become a necessary tool for agricultural production management and soil pollution prevention. Addressing the challenges posed by the complex terrain and fragmented land parcels in karst mountainous regions, as well as the frequent presence of cloudy and foggy weather conditions, the extraction efficacy of mulching films is compromised. This study utilized a DJI Mavic 2 Pro UAV to capture visible light images in an area with complex terrain features such as peaks and valleys. A plastic film sample dataset was constructed, and the U-Net deep learning model parameters integrated into ArcGIS Pro were continuously modified and optimized to achieve precise plastic film identification. The results are as follows: (1) Sample quantity significantly affects recognition performance. When the sample size is 800, the accuracy of plastic film extraction notably improves, with area accuracy reaching 91%, a patch quantity accuracy of 96.38%, and an IOU and F1-score of 85.89% and 94.20%, respectively, compared to the precision achieved with a sample size of 300; (2) Different learning rates, batch sizes, and iteration numbers have a certain impact on the training effectiveness of the U-Net model. The most suitable model parameters improved the training effectiveness, with the highest training accuracy achieved at a learning rate of 0.001, a batch size of 10, and 25 iterations; (3) Comparative experiments with the Support Vector Machine (SVM) model validate the suitability of U-Net model parameters and sample datasets for precise identification in rugged terrains with fragmented spatial distribution, particularly in karst mountainous regions. This underscores the applicability of the U-Net model in recognizing plastic film coverings in karst mountainous regions, offering valuable insights for agricultural environmental health assessment and green planting management in farmlands.

**Keywords:** karst mountainous terrain; UAV; U-Net modeling; ground cover recognition

## 1. Introduction

China, as a populous nation, considers food security a paramount strategic resource. Agricultural plastic films, utilized for ground cover, play a crucial role in increasing crop yields, enhancing soil temperature, reducing water evaporation, preventing pest attacks, and mitigating diseases caused by certain microorganisms. Consequently, they are widely applied in agricultural production [1]. However, with the increasing utilization of plastic films, the problem of plastic film residues has become increasingly severe. Effectively extracting plastic films has thus become a critical research problem. The dispersed nature, regional complexity, and diverse management of plastic film coverings make remote sensing technologies advantageous in plastic film identification. This is particularly essential for obtaining accurate spatiotemporal distribution information in China, aiding in farm environmental health assessment, plastic film recycling management, and supporting the implementation of low-carbon agriculture [2].

Currently, traditional methods for acquiring plastic film cover information primarily rely on labor-intensive and time-consuming manual field measurements, with challenges in ensuring data accuracy. As remote sensing data and information extraction technologies mature, the automated extraction of plastic film cover information has become more convenient. In recent years, the methods for identifying plastic films have gradually shifted from traditional manual field surveys to segmentation extraction based on remote sensing images. The detection of agricultural plastic film coverage using satellite remote sensing imagery has emerged as a research hotspot. Lu [3] proposed a threshold model based on moderate-resolution MODIS-NDVI time-series data, determining the threshold for detecting plastic film by analyzing data related to plastic film features. Xiong [4] presented a method for agricultural plastic film monitoring based on multisource remote sensing data, including steps such as plastic film information extraction, classification, and area estimation. Using high-spatial-resolution satellite imagery and spectral and texture features, they achieved the rapid detection and monitoring of plastic film cover over large areas. Chen [5] utilized high-spatial-resolution satellite imagery to develop a remote sensing index model for plastic greenhouses. By extensively exploring spectral and textural features and employing logistic regression analysis, they achieved the precise extraction of plastic film coverage in greenhouses. Picuno [6] analyzed the spatiotemporal distribution characteristics and extraction methods of plastic film cover in the landscape of southern Italy using remote sensing and object modeling techniques based on Landsat TM imagery. The results indicated widespread plastic film cover in southern Italy, and the plastic film extraction method significantly impacted landscape feature extraction.

Although satellite remote sensing has advantages such as wide coverage, abundant information, and freedom from ground restrictions, its limitations, such as long image acquisition cycles and susceptibility to cloudy and foggy weather in Guizhou, hinder plastic film identification accuracy, making it challenging to meet the identification needs of small cultivated areas and fragmented crop planting in karst mountainous regions.

With the continuous development of drone technology, utilizing drone imagery for information extraction has become a rather popular research direction. Drones offer advantages such as high maneuverability, high spatial resolution, and timeliness, adapting well to complex environments and exhibiting low costs. Currently, an increasing number of scholars are utilizing drones for detection purposes [7,8]. However, most identification methods based on drone imagery still rely on traditional satellite remote sensing interpretation methods, involving the manual selection of spectral, texture, and shape features for classification. This not only requires specialized domain knowledge but also entails significant computational efforts. Since Krizhevsky et al. [9] used deep learning technology to beat the world record in the ImageNet large-scale visual recognition competition, deep learning has opened up new prospects for applications in image classification, semantic segmentation, and other fields. Yang [10] extracted plastic film from high-resolution drone imagery using deep semantic segmentation technology. They established a convolutional neural network model to achieve the precise identification and classification of plastic film-covered areas. The results indicated that plastic film extraction based on deep semantic segmentation technology exhibited high accuracy and reliability, providing a new solution for monitoring and managing plastic film-covered areas. Sun [11] proposed a drone aerial monitoring method for greenhouses and plastic-covered farmland based on the SegNet deep semantic segmentation method, combining texture and spectral features. They used a convolutional neural network to extract plastic film and achieve the precise identification and classification of greenhouse and plastic film-covered areas. Zheng [12], comparing the effects of deep learning methods, U-Net methods, and Support Vector Machine (SVM) algorithms in extracting plastic film from greenhouses, constructed the ENVINet5 deep learning model to extract plastic film through semantic learning. Song [13] proposed using a pooling module to extract target features with a large receptive field based on a deep learning model and optimized the model by integrating high-level and low-level features.

In the aforementioned studies, the basic extraction of information regarding plastic film coverings in farmlands, including plastic film greenhouses, has been achieved. However, most of these studies focus on the remote sensing monitoring of large flat areas. Karst areas account for approximately 15% of the world's land area [14]. China has the largest and most widely distributed karst area [15], with the southwest bare karst region, centered in Guizhou, being the largest and most densely distributed area globally [16]. The rugged surface and extremely poor soil of karst terrain are unfavorable for agricultural development [17], leading to the popular saying in the Yun-Gui Plateau region: "No three flat miles, no three sunny days, no three taels of silver." Crop growth in karst mountainous regions is complex, with scattered planting distributions. Therefore, there is a need for more flexible, efficient, rapid, and accurate methods for plastic film recognition and monitoring in complex terrain areas.

With the continuous development of deep learning technology, its widespread application in automatic feature extraction and image fitting in the field of computer vision has provided new avenues for addressing target recognition issues in medium and high-resolution remote sensing images. In 2015, Ronneberger [18] proposed the U-Net model to tackle challenges in image segmentation. This model has shown outstanding performance in medical image segmentation, demonstrating strong generalization capability and excellent segmentation performance. As a result, it has become one of the most highly acclaimed classic models and has been widely applied in various fields. In land use classification of satellite remote sensing images, some scholars, such as Ulmas P. [19], utilized the U-Net model for land cover classification of high-resolution satellite images. Additionally, the U-Net method has been applied in building detection and road extraction in aerial images. For instance, Irwansyah E. [20] employed an improved U-Net model for building detection in urban aerial images, achieving an average training accuracy of 0.83. In the field of intelligent transportation, the U-Net model has been used for the real-time detection and tracking of vehicles and pedestrians as well as for road segmentation. Yang X. [21] utilized the U-Net model for vehicle detection and recognition in urban road images. Furthermore, the application of the U-Net model in agriculture is growing and includes crop growth monitoring and pest detection. Su Z. et al. [22] proposed an end-to-end, pixel-to-pixel rice lodging identification semantic segmentation method using an improved U-Net network model for unmanned aerial vehicle remote sensing images which achieved an accuracy of 97.30% and proved suitable for small sample datasets. In plastic film extraction, Zhai Z. et al. [23] combined unmanned aerial vehicle-acquired images of cotton fields with the U-Net model for image segmentation, achieving an average Mean Intersection over Union (MIOU) of 87.53%. Overall, as a deep learning method, the U-Net model has demonstrated significant potential and numerous application prospects in the fields of image segmentation and target recognition.

In this context, this study focuses on the Fengcong Dam area in Anlong County, Guizhou Province, China. Utilizing a DJI Mavic 2 Pro drone, a large number of visible light images covering farmland with plastic film were obtained. A plastic film sample dataset was constructed, and the U-Net deep learning model was trained to identify plastic film with the aim of extracting information on plastic film cover in fragmented terrain areas. This study aims to provide decision-making support for plastic film surveys, farm environmental health assessments, and green planting management in farmland, and to serve as a reference for the recognition and detection of agricultural plastic film under complex geographical conditions.

## 2. Materials and Methods

### 2.1. Study Area

The study area as shown in Figure 1 is located in Anlong County, Guizhou Province, which is situated in a transitional slope zone from the Yunnan–Guizhou Plateau to the hills of Guangxi. It is positioned at 105°20′ E and 25°14′ N. The area within the study site covers 0.196 km$^2$ and features significant terrain undulations, forming a well-developed karst

landscape, characteristic of a typical Fengcong Dam area. Anlong County experiences a subtropical monsoon humid climate with an average annual temperature of 15.3 °C and an average precipitation of 1256 mm. It is designated as a key grain production functional zone and an important agricultural product protection zone by the State Council. The cultivated land in the area is approximately 2 km² and is predominantly used for the cultivation of rice and vegetables. Additionally, there is a modern agricultural demonstration base covering approximately 1 km², featuring crops such as strawberries, cantaloupes, watermelons, asparagus, lotus roots, grapes, and prickly pear seedlings. Due to the influence of topography and weather systems, the research area is prone to regional meteorological disasters such as droughts and low-temperature frost. Plastic thin films are widely used here in response to these conditions.

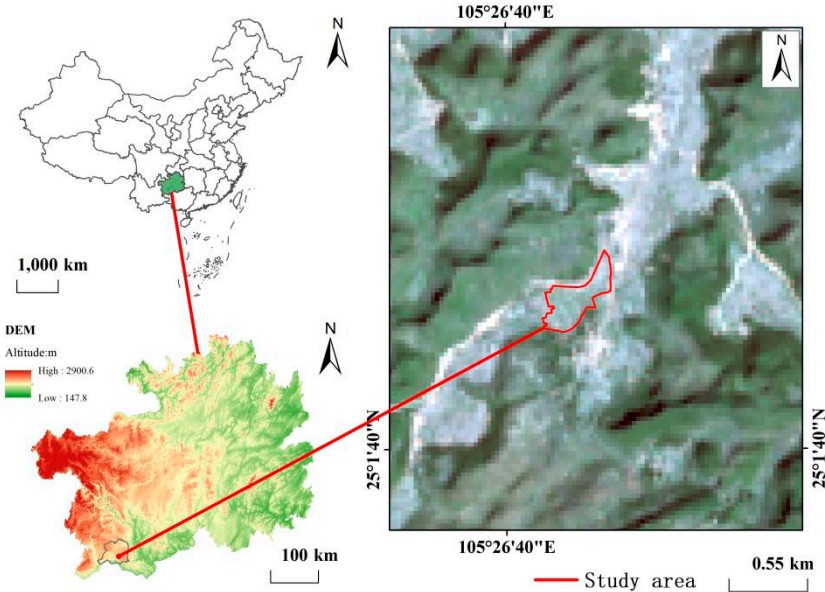

**Figure 1.** Location map of the study area.

## 2.2. Data Acquisition and Preprocessing

Clear or partly cloudy weather conditions were selected for data collection, taking into account variations in the solar incidence angle, which lead to the appearance of shadow areas in the fields. There are distinct differences in the color characteristics of plastic film and soil between shadow areas and areas directly illuminated by sunlight. Under ideal flight conditions, data collection was conducted during stable solar radiation intensity and clear, cloud-free sky conditions. This helped to avoid the loss of texture features in images due to cloud cover. Considering the often cloudy and foggy weather in the study area and the practical applicability of the method, image collection was performed on 21 February 2020, from 12:00 to 15:00, under overcast weather conditions with a wind speed of 2 m/s, meeting the safety conditions for drone operation. The drone used for the aerial survey was the DJI Mavic Pro v2.0 Professional Edition, and flight mission planning was executed using DJI GS Pro (Ground Station Pro) Professional edition with a fully automatic waypoint flight operation.

Upon surveying the flight task area, it was observed that high-voltage transmission lines crossed the dam area in a northeast–southwest direction. Additionally, there were high-power wireless signal stations for mobile operators such as China Telecom, China Mobile, and China Unicom on the mountain top, which could potentially impact the normal flight of the drone. To ensure flight safety, a flight altitude of 450 m was set. To guarantee image quality, a hover-and-capture method was employed with a heading overlap of 80% and a side overlap of 70%. The exposure mode was set to automatic exposure with ISO-100, a focal length of 10 mm, a maximum aperture of 2.97, an exposure time of 1/40 s, and a

flight speed of 9.5 m/s. The flight area covered 2.7 km². The images captured by the DJI Mavic 2 Pro were RGB true-color images. Ground control points were collected using RTK technology. The images were processed and stitched using Pix4Dmapper 4.4.12, including calibration processing (incorporating control points and control point encryption) followed by automated processing. This included initialization processing (setting the feature point image ratio to full high accuracy), point cloud encryption (image ratio of 2/1), and digital orthomosaic image (DOM) and digital surface model (DSM) generation (with a resolution of 1 times ground resolution). Finally, quality evaluation of the images was conducted through bundle block adjustment of details, internal camera parameters, ground control points, etc. The images were then enhanced, color-balanced, cropped, and reconstructed to obtain unmanned aerial vehicle (UAV) RGB remote sensing images with a spatial resolution of 0.1 m for the study area. As shown in Figure 2, area A represents the test zone images, while B represents the validation zone images.

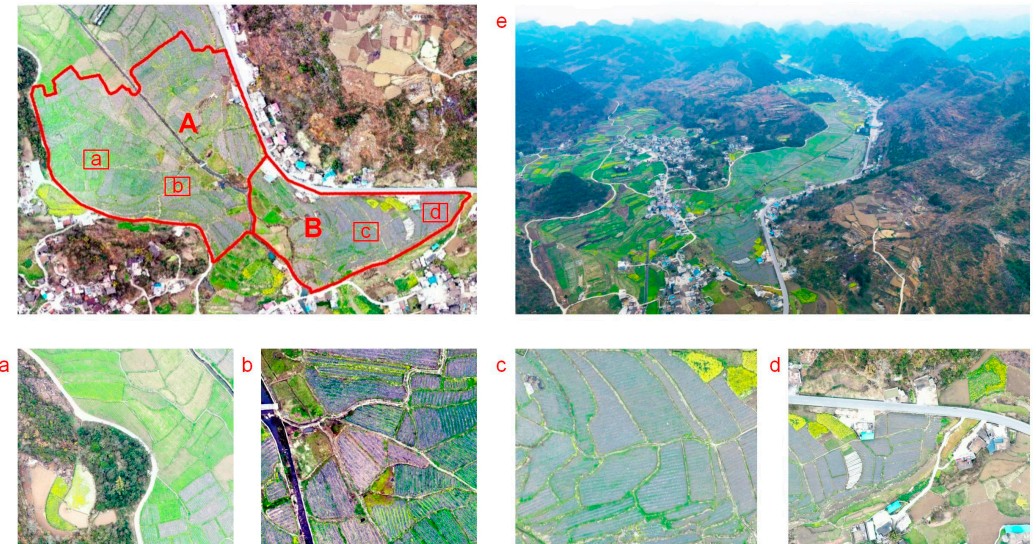

**Figure 2.** In the study area, section A is designated as the test area image, while section B serves as the validation area image. Images (**a**,**b**) represent enlarged views of the mulch in the test area, while images (**c**,**d**) depict corresponding enlarged views of the mulch in the validation area. Image (**e**) shows an oblique view of the study area.

### *2.3. Research Methods*

2.3.1. Technical Route

The workflow of the data processing, as illustrated in Figure 3, involved the following steps: firstly, we utilized an unmanned multi-rotor aircraft as a platform carrying a visible light sensor to collect images of land features. Subsequently, the images underwent processing steps such as calibration, stitching, and image enhancement. Secondly, employing ArcGIS Pro 3.0.1 in conjunction with field survey data, plastic film samples were selected through visual interpretation and field sampling. This process involved constructing a dataset of plastic film samples and analyzing the recognition performance concerning different sample quantities and background colors. Following this, parameter optimization for the U-Net model was performed, and the model's accuracy was evaluated to select the optimal configuration. Ultimately, the automated extraction of agricultural mulching information was achieved based on the recognized plastic film data.

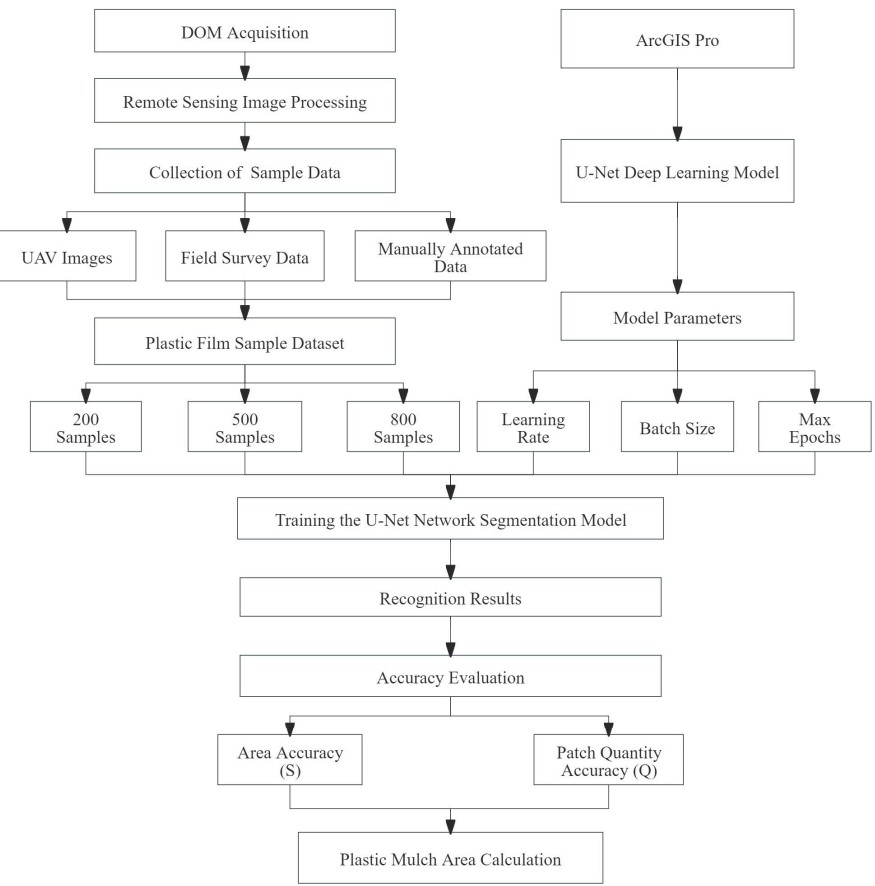

**Figure 3.** Roadmap.

### 2.3.2. The U-Net Model

The U-Net model is an improved and extended version of Fully Convolutional Networks (FCNs), designed as a semantic method. Its name originates from the "U" shape formed by its structure, and it was initially widely applied in the semantic segmentation of medical images [24,25]. As shown in Figure 4, the model consists of a contracting path on the left and an expansive path on the right. The contracting path follows a typical convolutional neural network structure, extracting features from input images and progressively down-sampling the resolution while increasing the number of features. This process captures features at different scales and abstraction levels, enabling the model to capture more semantic information. The expansive path, in contrast, involves an up-sampling process that restores the low-resolution feature maps to the original image resolution. Simultaneously, it extracts information from the feature maps to aid in the recovery of details and edges, thereby guaranteeing precise localization of segmentation positions for the target [26]. U-Net is a commonly used and relatively simple segmentation model in deep learning. It is easy to construct, efficient, and can be trained on small datasets. It exhibits significant advantages in image semantic segmentation, allowing for precise segmentation of images. It is particularly suitable for image segmentation in scenarios with complex backgrounds and fragmented targets [27,28]. In recent years, the U-Net model has found extensive applications in the field of agriculture, demonstrating significant potential in areas such as land feature recognition and detection.

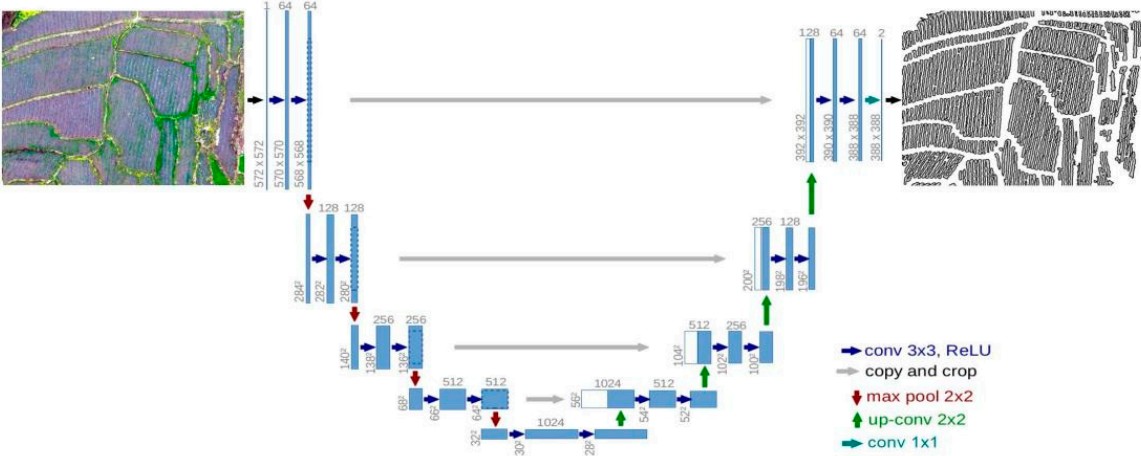

**Figure 4.** The U-Net model.

### 2.3.3. Training Environment

The software selected for image processing and analysis in this study was ArcGIS Pro. Additionally, the Tensorflow 2.7.0 framework, along with relevant deep learning libraries and tools, was installed to facilitate the construction and training of the U-Net model for the extraction of plastic film areas. During the training process of the U-Net neural network model, there are significant requirements for computational power, memory, and GPU memory. Therefore, the experimental setup carefully considered the demands of both software and hardware environments. High-performance hardware and professional software tools were chosen to ensure the efficiency of model training and the accuracy of results. The software and hardware environment used in the study is detailed in Table 1.

**Table 1.** Hardware and software environment.

| Baseline Configuration | Hardware Configuration | Software Configuration | Software Version |
|---|---|---|---|
| System (Microsoft Corporation, Washington, DC, USA) | Windows 10 Home Edition | CUDA | 10.1 |
| CPU (Microsoft Corporation, Washington, DC, USA) | Inter® Core (TM) i5-8265U | Python | 3.7 |
| Hard Disk (Seagate, Fremont, CA, USA) | 500 GB | Tensorflow | 2.7.0 |
| Graphics Card (NVIDIA Corporation, Santa Clara, CA, USA) | NVDIA GeForce MX250 | Keras | 2.7.0 |

### 2.3.4. Building the Dataset

Due to the rugged terrain of karst mountainous areas, the crop planting areas are relatively small, resulting in a fragmented spatial distribution of plastic film. The length and shape of the plastic film may vary due to factors such as terrain, vegetation cover, and land use. These inconsistencies may result in various sizes and shapes of plastic film fragments in the images, thereby increasing the complexity of extraction algorithms. Additionally, the shape of the plastic film may vary due to factors such as crop type and agricultural practices and may include straight, curved, and meandering shapes. This diversity of shapes may pose challenges for extraction algorithms in identifying and segmenting plastic film, especially for meandering or curved shapes, which may lead to instances of omission or misidentification by the extraction algorithm. Therefore, the area and shape of the plastic film vary due to geographical conditions and geological processes, exhibiting diversity and complexity. Based on the background characteristics and distribution of plastic film in the study area, we selected four representative sample images, as shown in Table 2, including the first image (I) and the second, third, and fourth images (II, III, IV). These sample images exhibit diverse patterns of plastic film

distribution and backgrounds, representing the variability in the research area. In the first image (I), plastic film is mainly concentrated in farmland and is surrounded by buildings, roads, and other vegetation. Plastic film distribution is uniform in images II, III, and IV, but image II has a more complex background with a higher presence of weeds. Images III and IV serve as validation images. Using manual annotation in ArcGIS pro, the plastic film areas were delineated to create plastic film labels. Finally, the annotated plastic film regions were segmented into 224 × 224-pixel patches for subsequent training and model validation. This process resulted in a dataset comprising 800 samples. To enhance the model's generalization capability and accuracy, we divided the dataset into three sets of samples for training, comprising 300, 500, and 800 samples, respectively. This approach allowed for a gradual increase in the amount of data during the training process, thereby improving the performance and accuracy of the model. We evaluated the model's performance and accuracy by comparing it with manually delineated patches (equivalent to real patches) using a validation method.

**Table 2.** True-color images and manually annotated maps.

| Serial Number | True-Color Image | Manual Labeling |
|:---:|:---:|:---:|
| I |  |  |
| II |  |  |
| III |  |  |

**Table 2.** *Cont.*

| Serial Number | True-Color Image | Manual Labeling |
|:---:|:---:|:---:|
| IV | 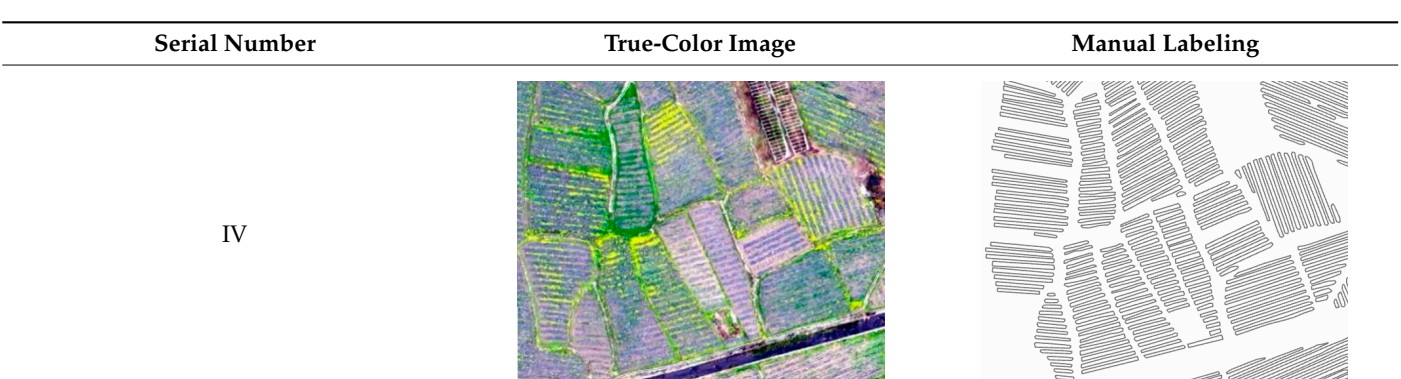 | |

### 2.3.5. Fine-Tuning of Parameters

Learning rate [29] is a critical parameter adjustment in neural network training as it controls the step size during each iteration. It determines the extent of parameter adjustment with each gradient update, indicating the rate at which existing information is overwritten by new information. The choice of learning rate directly influences the model's convergence speed and training effectiveness. An excessively large learning rate may lead to increased oscillation during training, preventing convergence to an optimal state. Conversely, a too-small learning rate may slow down training speed or even result in stagnation. Therefore, selecting an appropriate learning rate is crucial. In this study, under fixed batch size and other parameters, learning rates of 0.05, 0.01, 0.005, and 0.001 were employed for model training to identify the optimal learning rate.

Batch size [30] refers to the number of training samples processed in a single iteration of U-Net model training. It determines the amount of data the model processes in each iteration, significantly impacting training and performance. Increasing batch size generally allows the model to better utilize training data, enhancing performance and accuracy. When the batch size becomes too large, memory usage significantly increases. Therefore, selecting an appropriate batch size requires balancing performance and memory usage. In this experiment, with a fixed learning rate and other parameters, batch sizes of 8, 10, and 12 were used for model training. By comparing the performance of the model under different batch sizes, the most suitable batch size for the current model could be determined to achieve better training results and higher prediction accuracy.

Max epochs [31] represent the maximum number of iterations for U-Net model training. One epoch refers to one pass of the dataset through the neural network, involving both forward and backward passes. As the number of epochs increases, the model gradually learns more data features, improving prediction accuracy. However, more epochs do not necessarily lead to better outcomes, as an excessive number of iterations can significantly increase the model's runtime and pose a risk of overfitting. In this experiment, with a fixed learning rate and batch size, different max epochs numbers of 15, 20, 25, and 30 were used for model training. By comparing the model's performance under different max epochs, the most suitable max epochs for the current model could be determined, balancing efficiency and effectiveness.

### 2.3.6. Accuracy Evaluation

To quantitatively evaluate the performance of the model, we employed four accuracy evaluation metrics: area extraction accuracy, object count accuracy [32], Intersection over Union (IOU), and F1-score [33]. The actual area of the plastic film was obtained through field measurements and manual delineation on visible light images captured by the unmanned aerial vehicle (UAV). Subsequently, we calculated the area extraction accuracy and patch count error of the model. IOU is used to evaluate the accuracy of plastic film segmentation, calculated as the ratio of the intersection area between the predicted seg-

mentation result and the ground truth segmentation result to the union area. A higher IOU value indicates greater consistency between the predicted and ground truth results, reflecting the accuracy of segmentation [34]. On the other hand, F1-score comprehensively considers both precision and recall, providing a comprehensive assessment of the overall performance of the model [35]. By comparing these metrics with the actual plastic film area, we could assess the model's performance. The calculation formulas are as follows:

$$S = (1 - |(S_1 - S_0)/S_0|) \times 100 \tag{1}$$

$$Q = (1 - (Q_1 - Q_0)/Q_0) \times 100 \tag{2}$$

$$\text{IOU} = \frac{TP}{TP + FN + FP} \tag{3}$$

$$\text{Precision} = \frac{TP}{TP + FP} \tag{4}$$

$$\text{Recall} = \frac{TP}{TP + FN} \tag{5}$$

$$\text{F1-score} = \frac{2 \times \text{Precision} \times \text{Recall}}{\text{Precision} + \text{Recall}} \tag{6}$$

In these equations, *S* represents the accuracy of area extraction, where $S_1$ is the extracted area of the plastic film, and $S_0$ is the reference area. *Q* denotes the accuracy of patch count, with $Q_1$ being the extracted number of patches and $Q_0$ being the actual number of plastic film patches. *TP* represents correctly identified plastic film areas, *FP* represents erroneously identified plastic film areas, and *FN* represents missed plastic film areas.

## 3. Result Analysis

### 3.1. Analysis of Optimal Parameters

This study employed ArcGIS Pro software as the experimental platform and utilized the Keras deep learning framework to construct the experimental environment for model training and parameter optimization. As the backbone model, we selected NesNet-34, which has demonstrated outstanding performance and stability in image classification and feature extraction. During the experiments, the initial learning rate was set to 0.01, batch size was set to 10, and the maximum number of iterations was limited to 25. In the experimental process, we attempted different learning rate settings, including 0.01, 0.05, 0.001, and 0.005. We found that different learning rates had varying effects on the accuracy of area prediction and patch number prediction for the sample dataset. The variation curves are illustrated in Figure 5. Through analysis and comparison, we determined that the model achieved optimal classification performance when the learning rate was set to 0.001. Therefore, we uniformly set the learning rate parameter to 0.001 in subsequent experiments.

In this experiment, we investigated the impact of different batch sizes (8, 10, 12) on the accuracy of plastic film recognition. Considering the limitations of GPU memory, dataset capacity, and input image size in the experimental environment, the maximum batch size supported by this study was set to 12. The evaluation metrics for plastic film recognition accuracy with different batch sizes are presented in Figure 6. The experimental results indicate that when the batch size is set to 10 and 12, there is minimal variation in the evaluation metrics. Moreover, as the batch size increases, the oscillation during the training process gradually decreases, and memory utilization also improves. Therefore, for this experiment, a batch size of 10 was ultimately selected.

To determine the appropriate number of iterations, this experiment, under the premise of fixed learning rate, batch size, and other parameters, trained the model with different numbers of iterations (15, 20, 25, 30). The maximum number of iterations needs to be adjusted based on the resources and time available for machine learning. The impact of different numbers of iterations on accuracy is shown in the Figure 7. The segmentation performance of the model was optimal when the number of iterations was 25.

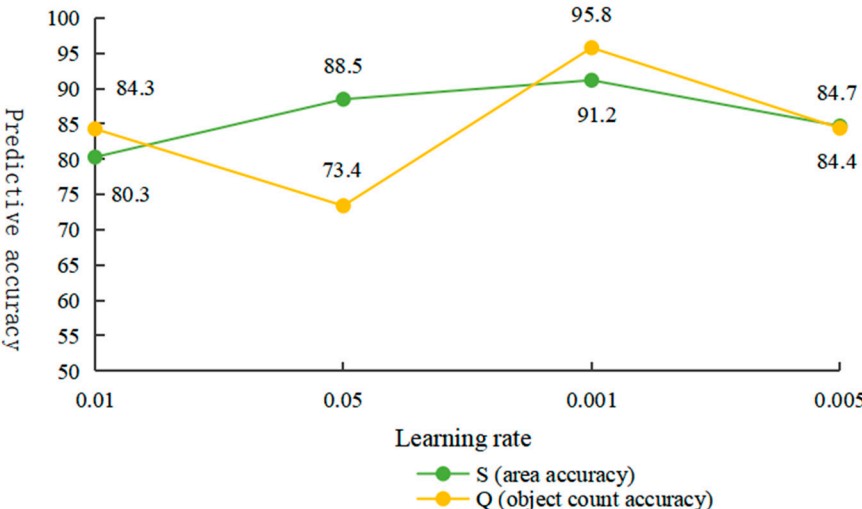

**Figure 5.** Learning rate optimization.

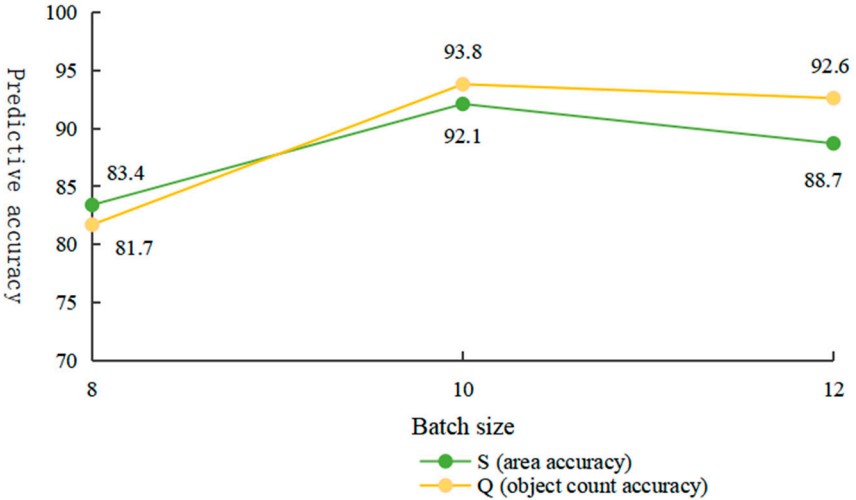

**Figure 6.** Batch size optimization.

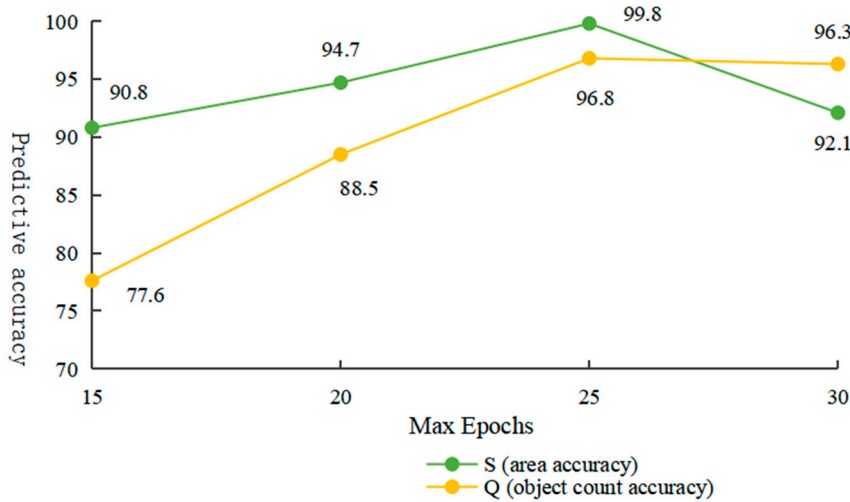

**Figure 7.** Optimization of max epochs.

### 3.2. Analysis of Recognition Results

3.2.1. Accuracy Evaluation

According to the U-Net Model's results for UAV remote sensing image recognition, as shown in Table 3, the model achieved the best training results when the learning rate was 0.001, the batch size was 10, and the number of iterations was 25. Segmentation accuracy is detailed in the table below. The results indicate that as the number of samples in the dataset changes, the accuracy of the recognition results also varies accordingly. With a larger number of samples, recognition performance improves. Notably, with 800 samples, area accuracy is 91%, object count accuracy is 96.38%, and the IOU and F1-score are 85.89% and 94.20%, respectively, representing the highest performance. Next are 500 samples, with an area accuracy of 79.93%, a patch quantity accuracy of 89.61%, and an IOU and F1-score of 83.18% and 90.81%, respectively. The least effective are 300 samples, with an area accuracy of 66.9%, a patch quantity accuracy of 78.78%, and an IOU and F1-score of 89.04% and 94.20%, respectively. With the increase in sample size, area accuracy improved by 24.1%, patch quantity accuracy by 17.6%, and IOU and F1-score by 13.76% and 8.31%, respectively. As shown in the table, increasing the sample dataset helps improve the overall performance and extraction accuracy of the U-Net model.

**Table 3.** Accuracy of image classification results.

| Sample Size | 300 | 500 | 800 |
|---|---|---|---|
| Area Accuracy (S) | 66.90 | 79.93 | 91.00 |
| Object Count Accuracy (Q) | 78.78 | 89.61 | 96.38 |
| IOU | 75.28 | 83.18 | 89.04 |
| F1-score | 85.89 | 90.81 | 94.20 |

3.2.2. Visual Analysis

Table 4 presents the identification results under fixed learning rate, batch size, iteration times, and sample size conditions. Red contours represent identified plastic film patches. Observing the identification results for different sample sizes reveals that with an increase in sample size, the completeness of plastic film extraction improves, corresponding to enhanced identification accuracy and improved fragmentation of patches. Influenced by factors such as surrounding vegetation, segmentation becomes more challenging. However, by increasing the sample size, we can effectively address this issue, bringing the number of patches closer to the sample size, thereby enhancing patch quantity accuracy and improving the accuracy of plastic film identification. Therefore, from the identification results, we conclude that in dealing with complex situations, a greater number and greater diversity of samples lead to better identification results and higher accuracy.

**Table 4.** Recognition results of U-Net model trained with different samples.

| Sample Size | Scenario 1 | Scenario 2 | Scenario 3 | Scenario 4 |
|---|---|---|---|---|
| 300 | | | | |

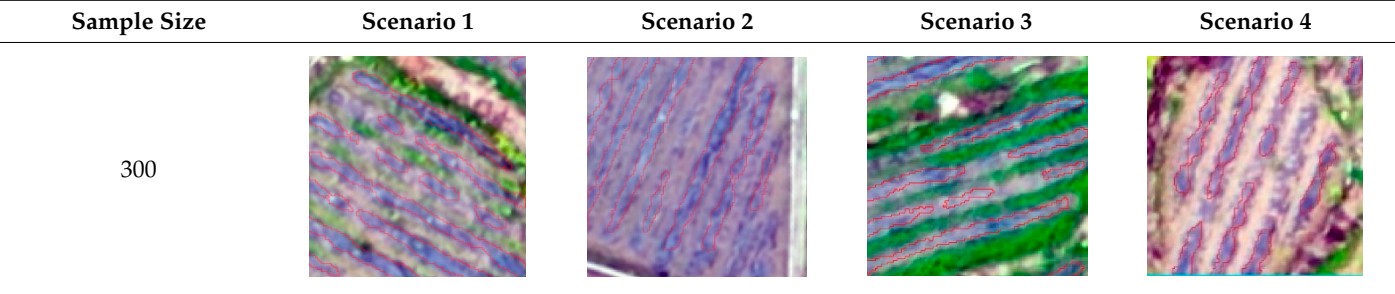

**Table 4.** *Cont.*

| Sample Size | Scenario 1 | Scenario 2 | Scenario 3 | Scenario 4 |
|:---:|:---:|:---:|:---:|:---:|
| 500 |  |  |  |  |
| 800 |  |  |  |  |

### 3.3. Comparative Analysis of Methods

To highlight the effectiveness of the U-Net model employed in plastic film recognition in this study, we conducted comparative experiments with the traditional Support Vector Machine (SVM) method [36], using an analysis of 800 plastic film samples. For the SVM recognition results, we conducted the same accuracy evaluations. As indicated by the data in Table 5, compared to the SVM model, the U-Net model demonstrated improvements of 1.10% in area accuracy, 20.42% in patch quantity accuracy, 9.94% in IOU, and 5.87% in F1-score, respectively. Through comparative analysis, we showcased the superiority of the U-Net model over the SVM method in plastic film recognition accuracy, with the U-Net model being able to identify plastic film regions more accurately and efficiently. This further validates the application potential of the U-Net model in plastic film recognition and underscores its superiority in addressing complex terrain and land cover conditions.

**Table 5.** Precision comparison of different models.

| Models | U-Net | SVM |
|:---:|:---:|:---:|
| Area Accuracy (S) | 91.00 | 89.90 |
| Object Count Accuracy (Q) | 96.38 | 75.96 |
| IOU | 89.04 | 79.10 |
| F1-score | 94.20 | 88.33 |

In visual analysis, we further observed that the patches generated by the U-Net model for plastic film recognition are more complete and coherent. As shown in Table 6, compared to the traditional SVM method, the plastic film patches generated by the U-Net model exhibit greater spatial consistency and morphological continuity, enabling a more accurate reflection of the actual plastic film coverage. In comparative experiments, we found that the SVM method tends to produce broken, fragmented, or omitted plastic film patches, while the U-Net model better preserves the continuity and integrity of the plastic film. These visual analysis results further validate the superiority of the U-Net model in plastic film recognition.

**Table 6.** Different model recognition effects.

| Models | Scenario 1 | Scenario 2 | Scenario 3 |
|---|---|---|---|
| U-Net |  |  |  |
| SVM |  |  |  |

## 4. Discussion

### 4.1. Applicability of the Method

This study utilized unmanned aerial vehicle (UAV)-based visible light imagery to explore the application of the U-Net model in monitoring plastic film coverage in high-altitude mountainous farmland and investigated its suitability for such tasks. The rapid development of UAV technology, characterized by high mobility, low cost, and enhanced safety, provides a new avenue for geographical information acquisition. Traditional monitoring methods face limitations due to the diverse land cover types and complex terrain of karst mountainous farmlands. Therefore, this research introduces multi-rotor UAVs to monitor plastic film coverings in high-altitude mountainous areas, investigating their advantages and effectiveness in practical applications. Our results indicate that multi-rotor UAVs exhibit high cost-effectiveness and safety, coupled with unique advantages in land cover monitoring. They can swiftly acquire high-resolution visible light images, effectively monitoring features in fragmented planting areas. Moreover, visible light images captured by drones can also reflect plastic film coverage under different backgrounds, providing decision-making support for plastic film surveys, farmland health assessments, and modern agricultural park management. Additionally, determining the optimal sample size is a complex issue involving multiple factors, including processing time, machine resources, data quality, and model complexity. Through experimentation and comparison with different sample sizes, we found that with a sample size of 800, the accuracy of plastic film recognition can be effectively improved, demonstrating good performance during the training process. This result also provides an effective method for determining the optimal sample size for subsequent research. Furthermore, by using cross-validation techniques, we comprehensively evaluated the model's performance under different sample sizes, thereby providing reliable evidence for selecting the optimal sample size. These methods not only enhance the training efficiency and performance of the model but also effectively save time and resource costs, providing strong support for subsequent research and practical applications.

### 4.2. Differences from Existing Research

This study, conducted in a karst mountainous region of southern China, utilized a UAV remote sensing platform, specifically the DJI Mavic 2 Pro, to extract information about plastic film coverings in complex habitats. This approach effectively addresses challenges in obtaining high-quality remote sensing image data for crop information extraction in the fragmented and environmentally fragile karst mountainous terrain with frequent cloudy and foggy weather conditions. In contrast, previous research employing medium-

and low-resolution satellite remote sensing images, such as Landsat TM and Landsat 8, combined spectral and texture features to improve classification accuracy. For instance, Lu [37] achieved overall accuracy rates of 85.27% and 95% using Landsat-5 TM images, and Hasituya [38] achieved an overall classification accuracy of up to 94% by combining spectral and texture features based on Landsat-8 remote sensing data. In this study, the U-Net model was employed for the semantic segmentation of drone images, achieving a patch count of 96.38%, an area accuracy of 91%, and an IOU and F1-score of 85.89% and 94.20% respectively. In comparison with the studies conducted by Lu and Hasituya, our research may exhibit differences, which could stem from variations in data sources, study areas, methodologies, algorithms, and parameter settings. However, our study, tailored to specific geographical conditions and application needs, has devised methods and algorithms better suited for monitoring agricultural mulching in karst mountain areas. Consequently, our research findings remain somewhat comparable, potentially demonstrating superior applicability and efficacy in particular application scenarios.

*4.3. Limitations*

In future research, it is imperative to critically reflect on and address the limitations of the current study to further enhance the quality and reliability of our research outcomes. Throughout our investigation, the restricted endurance of the unmanned aerial vehicle (UAV) constrained our research to relatively small areas, thereby limiting our capacity for extensive data collection over larger regions. In subsequent studies, employing UAVs with higher endurance capabilities or optimizing flight path planning algorithms may enable coverage and data collection over larger areas. Additionally, despite augmenting sample sizes and incorporating samples with diverse backgrounds, the issue of plastic film misidentification between roads still remains unresolved. Future endeavors will delve deeper into understanding the impact of various scenarios on plastic film extraction and endeavor to refine training strategies and parameter settings to bolster the model's recognition capabilities and accuracy in complex scenarios. Furthermore, while our study explored the influence of sample size on plastic film extraction, samples may still be susceptible to environmental factors such as illumination, weather, and ground reflectance. To comprehensively assess model performance, further investigation into the effects of these environmental factors on plastic film extraction is warranted. Efforts will be made to incorporate these factors into model training and optimization processes to enhance the model's robustness and reliability across diverse environmental conditions.

Future endeavors may continue to refine and advance plastic film recognition technology based on UAV visible light imagery. The exploration of high-resolution remote sensing data and the utilization of high-resolution orbital images for plastic film recognition and monitoring can enhance recognition accuracy and spatial resolution. Improvements in and optimizations of plastic film recognition algorithms and models aim to enhance recognition efficiency. Additionally, leveraging multi-temporal remote sensing image data for the temporal monitoring and change analysis of plastic film will facilitate a better understanding of the growth evolution patterns of plastic film and the impacts of agricultural management practices.

**5. Conclusions**

Considering the varied backgrounds of plastic film environments, we employed drones to swiftly capture high-resolution visible light images in a karst mountainous area. Simultaneously, the experimental zone was partitioned into four distinct areas. Utilizing the U-Net model with different parameters, such as learning rates, batch sizes, and iteration counts, we systematically compared the impact of these model parameters. After assessing the effects, the optimal training parameters were identified. Furthermore, we compared the recognition outcomes with varying sample quantities. Ultimately, the U-Net model was used for image segmentation to extract plastic film, and the area method was employed for

plastic film area calculation. This facilitated the swift identification and area calculation of plastic film, leading to the following key conclusions:

### 5.1. Deep Learning Framework and Parameter Optimization

Leveraging the U-Net model within a deep learning framework, this study extracted plastic film areas from UAV-based visible light images. Exploring various learning rates, batch sizes, and iteration counts, this study identified optimal model parameters to enhance training effectiveness and improve plastic film extraction accuracy. The best recognition accuracy was achieved with a learning rate of 0.001 (91.37%), batch size of 10 (92.14%), and iteration count of 25 (99.84%). Therefore, for UAV image-based plastic film extraction, the optimal parameter values for learning rate, batch size, and iteration count are 0.001, 10, and 25, respectively.

### 5.2. Validation of U-Net Model in Karst Highland Terrain

This study employed a U-Net model based on UAV visible light imagery for plastic film extraction and conducted comparative experiments with the traditional Support Vector Machine (SVM) method. By increasing the sample size, we effectively improved the training performance of the U-Net model, consequently enhancing the accuracy of plastic film identification. With a sample size of 800, the U-Net model demonstrated an area accuracy of 91%, a patch quantity accuracy of 96.38%, an IOU of 85.89%, and an F1-score of 94.20%. During training, there was a 24.1% increase in area accuracy, a 17.6% increase in patch quantity accuracy, and improvements of 13.76% and 8.31% in IOU and F1-score, respectively. These results validate the superiority of the U-Net model in plastic film identification. A comparative analysis of experimental results revealed that compared to the SVM method, the U-Net model exhibited higher area accuracy (increased by 1.10%), patch quantity accuracy (increased by 20.42%), IOU (increased by 9.94%), and F1-score (increased by 5.87%) in plastic film identification. These data further confirm the excellent performance of the U-Net model in plastic film identification and provide important reference for future optimization of model training and enhancement of plastic film identification effectiveness.

### 5.3. UAV Remote Sensing in Small-Scale Crop Recognition

In remote sensing identification studies in fragmented and small-scale agricultural geospatial contexts, UAV remote sensing holds vast application prospects and is poised to become an indispensable means of aerial remote sensing. This study explores the applicability of UAV visible light images in detecting plastic film mulch (PFM) in a karst mountainous area, considering the region's characteristics of cloudy and misty weather, fragmented crop planting areas, and strong PFM heterogeneity. The proposed method features ease of operation, automation, and high accuracy, meeting the requirements for PFM detection in fragmented terrains, thus boasting broad application prospects. Moreover, by extracting and identifying agricultural PFM, accurate calculations of the covered area and distribution of PFM can be obtained, providing methodological references for PFM recycling and management. Additionally, by identifying the areas covered by PFM, the crop planting area in the region can be inferred, thereby offering data support for agricultural production. The selected parameters of the U-Net model and the sample dataset in this research meet the requirements for precise PFM identification in karst mountainous areas characterized by significant terrain undulations and fragmented spatial distribution of cultivation. This validates the applicability of the U-Net model in PFM identification in karst mountainous areas. Furthermore, this method can assist in monitoring land use and understanding specific land utilization patterns and occupancy situations, thereby providing a scientific basis for land resource management and planning and offering research methods and a decision-making basis for agricultural environmental health assessment and green planting management in agricultural fields.

**Author Contributions:** All authors contributed to the manuscript. Conceptualization, X.D. (Xiaoyi Du); methodology, X.D. (Xiaoyi Du); validation, X.D. (Xiaoyi Du) and D.H.; formal analysis, X.D. (Xiaoyi Du); data curation, X.D. (Xiaoyi Du) and D.H.; writing—original draft preparation, X.D. (Xiaoyi Du); writing—review and editing, X.D. (Xiaoyi Du), D.H., L.D. and X.D. (Xiandan Du); visualization, X.D. (Xiaoyi Du); project administration, D.H.; funding acquisition, D.H. All authors have read and agreed to the published version of the manuscript.

**Funding:** This work was supported by the Guizhou Provincial Key Technology R&D Program (Qiankehe [2023] General No.211) and the Guizhou Provincial Basic Research Program (Natural Science) (Qiankehe [2021] General No.194).

**Institutional Review Board Statement:** Not applicable.

**Informed Consent Statement:** Not applicable.

**Data Availability Statement:** Data are contained within the article.

**Conflicts of Interest:** The authors declare no conflicts of interest.

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
