# Peer review of "Recognition of Plastic Film in Terrain-Fragmented Areas Based on Drone Visible Light Images"

_agriculture, doi:10.3390/agriculture14050736_

Round 1

Reviewer 1 Report

Comments and Suggestions for Authors

The article presents the development of a practical application with a high potential for transferability of the approach and method to other cases where it is necessary to identify and classify small objects in fragmented landscapes.

Analysing the article shows the need for:

i) to make explicit the interest of the work for locations other than China,

ii) to improve the bibliographical review and analysis of the framework of the methods, but in particular of similar or convergent examples of application of the U NET method;

iii) reinforce the analysis/depth of the results (present the basis for learning) and their interpretation and potential application (e.g. for generating agricultural statistics, land use/occupation statistics,...);

iv) present ways of processing and geometric and radiometric correction of aerial images obtained by drone

v) better explore analytically and graphically the ideal learning ratio and number of batches;

v) better explain the form/method of the real plastic area on the ground; this value influences all the final performance metrics;

vi) explain the apparent differences between the visual analysis of the images included in table 4 and the result of the object count accuracy (96.38%)!!!.

The article presents a relevant scope and interest, an interesting methodology with a high potential for transferability, but it lacks a greater/better bibliographical review and greater statistical/graphical exploration, as well as the English language and the framing of the results.

Comments on the Quality of English Language

Review  

Reviewer 2 Report

Comments and Suggestions for Authors

Dear Authors,

       The manuscript "Recognition of Plastic Film in Terrain-Fragmented Areas Based on Drone Visible Light Imagery" addresses a topic pertinent to the development of a sustainable agricultural practice.

       The type of target and its occurrence in the field are unclear. In chapter one you mention greenhouses. It would be interesting to present the occurrence of the target in the field.

       One of the differentials of the study would be its use in areas with rolling terrain, and by analyzing Figure 2 you can see that the agricultural areas are located in places with flat terrain.

       I would like to draw your attention to the following questions:

1) Please revise the abstract and make more evident the main contribution of the research and highlight place of study/country. All the conclusions are presented and one of them is highlighted.

2) Line 77: "exhibiting low costs". Depends on the category of UAV and the sensors used.

3) Lines 111 to 116: highlight the objectives of the study. Study site and UAV can be characterized in the appropriate chapter.

4) Line 160: present the configurations used to process the aerial images.

5) Line 162: "Sample Area a represents the test area image, while B represents the validation area image." Highlight these areas in Figure 2.

6) Chapter 2.2: which GNSS positioning method is used? RTK? Please elaborate on this.

7) Figure 2: would it be possible to add an image of the target (plastic film) taken in the field?

8) Line 238: "Cross-validation was employed to assess the model's performance and accuracy". The cross-validation technique is more appropriate for a regression process rather than classification. In this case, you get true positive, false positive, false negative. Review.

9) Equations 1 and 2: metrics such as accuracy, precision, recall and F1 are normally used in a classification process. The choice of this evaluation method does not coincide with the methods traditionally recommended for evaluating classification. The development of a weighted confusion matrix would provide other interesting metrics.

10) In order to give greater support to the results obtained, it is necessary to compare them with other studies or even carry out classification via SVM or RF, for example. Another alternative would be to compare the results obtained with other network models such as P Net.

11) The discussion chapter should be revised and based on the results obtained and not on aspects that have not been scientifically evaluated.

12) Adapt the conclusions so that they respond adequately to the objectives of the study.

13) Recommendations for future studies? With the spatial resolution obtained, it is possible to test the use of the methodology for orbital images with high spatial resolution in a new study.

I end my review by congratulating you on your study.

Respectfully,

Comments on the Quality of English Language

I request a revision of the wording of the manuscript, including punctuation. When introducing acronyms in the text for the first time, their meaning should be presented. Replace the term drone with one of the acronyms: UAV (unmanned aerial vehicle) or RPA (Remotely Piloted Aircraft System). The latter is the technical and internationally standardized term used by the International Civil Aviation Organization (ICAO) to refer to remotely piloted aircraft systems used for non-recreational purposes. The acronym UAV is still used in scientific publications and its use is more widespread. Avoid using long paragraphs as they tend to make reading confusing. I have marked in the comments of the digital file the places where the wording should be improved.

Author Response

Thank you very much for your review and suggestions on our paper. We have carefully read your comments and revised them in the paper. Please refer to the attachment for answers to specific questions.

Reviewer 3 Report

Comments and Suggestions for Authors

This paper uses drone imagery and deep learning to present an effective method for recognising plastic film in agricultural fields in terrain-fragmented karst mountainous regions. The study area in Anlong County, Guizhou, features undulating karst terrain, creating a fragmented spatial distribution of planting areas.

To address this, the researchers utilised a DJI Mavic 2 Pro drone to capture high-resolution visible light images covering approximately 2.7 square kilometres of farmland. They then manually annotated plastic film regions in four representative images showing different distribution patterns and backgrounds to create a plastic film sample dataset.

Various model parameters like learning rate, batch size, and number of iterations were tested to determine optimal values for training effectiveness.

Results found that the model achieved the best performance with a learning rate of 0.001, batch size of 10, and 25 iterations. Using this configuration, area extraction accuracy reached 91%, and patch quantity accuracy was 96.38% when trained on a dataset of 800 samples, demonstrating the model's high recognition ability.

I have read through the manuscript two times and did not find any major remarks to point out. The methodology and experiments are clearly described, and the results support the conclusions.

Minor comments are as follows:

Figure 4: image resolution is low and blurred. The authors could improve this figure by providing a higher-resolution version.

References: The reference list has 25 citations, a bit shorter than usual. Given that this is a technical paper on a specialised topic, including a few more relevant studies would help situate their work within the landscape of prior research.

- Object-based image analysis (OBIA): You're correct that OBIA is not mentioned, even briefly, in the introduction. Since OBIA is a common remote sensing approach for extracting land features, acknowledging it early on could provide more context for their choice to use a deep learning-based method instead.

Author Response

Thank you very much for your review of our paper. Please refer to the attachment for answers to specific questions.

Reviewer 4 Report

Comments and Suggestions for Authors

1/ L133: Could the authors add the study area boundary in the red box in Figure 1? In addition, please provide the total area (km2) of the study area.

2/ Figures 5 and 6: Please provide details of S and Q in the figures’ caption.

3/ L333-335 and L340: Do we have any approach to define the optimal number of sample size considering the processing time, machine resources, etc.? Please provide at least a comment here to highlight your suggestion(s) adopted from your model experience.

4/ Table 4: Could the author explain if we have a higher-resolution drone which can then capture higher-resolution images, do we need to increase the sample size to obtain similar or better results compared to the current results (in your work)?

5/ Section 4.2: Please provide more details why we have these differences compared to studies by Lu. [24] and Hasituya. [25]…

Author Response

(The authors gave the same response as above.)

Round 2

Reviewer 2 Report

Comments and Suggestions for Authors

Dear Authors,

       The manuscript “Recognition of Plastic Film in Terrain-Fragmented Areas Based on Drone Visible Light Imagery” The paper discusses the use of UAV and deep learning techniques for the identification and area calculation of plastic film in agricultural fields.

      I've read through the new version of your manuscript and can see that you've made a number of changes. This made the manuscript easier to read and the experiment clearer to understand.

      However, when I checked the questions, I saw that although you mention the second question about the term low costs in the cover letter. Please revise the manuscript and remove as answered in the cover letter.

Cordially,
